# Generating steganographic images via adversarial training

**Jamie Hayes**
University College London
j.hayes@cs.ucl.ac.uk

**George Danezis**
University College London
The Alan Turing Institute
g.danezis@ucl.ac.uk

## Abstract

Adversarial training has proved to be competitive against supervised learning methods on computer vision tasks. However, studies have mainly been confined to generative tasks such as image synthesis. In this paper, we apply adversarial training techniques to the discriminative task of learning a steganographic algorithm. Steganography is a collection of techniques for concealing the existence of information by embedding it within a non-secret medium, such as cover texts or images. We show that adversarial training can produce robust steganographic techniques: our unsupervised training scheme produces a steganographic algorithm that competes with state-of-the-art steganographic techniques. We also show that supervised training of our adversarial model produces a robust steganalyzer, which performs the discriminative task of deciding if an image contains secret information. We define a game between three parties, Alice, Bob and Eve, in order to simultaneously train both a steganographic algorithm and a steganalyzer. Alice and Bob attempt to communicate a secret message contained within an image, while Eve eavesdrops on their conversation and attempts to determine if secret information is embedded within the image. We represent Alice, Bob and Eve by neural networks, and validate our scheme on two independent image datasets, showing our novel method of studying steganographic problems is surprisingly competitive against established steganographic techniques.

## 1   Introduction

Steganography and cryptography both provide methods for secret communication. *Authenticity* and *integrity* of communications are central aims of modern cryptography. However, traditional cryptographic schemes do not aim to hide the presence of secret communications. Steganography conceals the presence of a message by embedding it within a communication the adversary does not deem suspicious. Recent details of mass surveillance programs have shown that meta-data of communications can lead to devastating privacy leakages[1]. NSA officials have stated that they "kill people based on meta-data" [8]; the mere presence of a secret communication can have life or death consequences even if the content is not known. Concealing both the content *as well as* the presence of a message is necessary for privacy sensitive communication.

Steganographic algorithms are designed to hide information within a *cover* message such that the cover message appears unaltered to an external adversary. A great deal of effort is afforded to designing steganographic algorithms that minimize the perturbations within a cover message when a secret message is embedded within, while allowing for recovery of the secret message. In this work we ask if a steganographic algorithm can be learned in an unsupervised manner, without

human domain knowledge. Note that steganography only aims to hide the presence of a message. Thus, it is nearly always the case that the message is encrypted prior to embedding using a standard cryptographic scheme; the embedded message is therefore indistinguishable from a random string. The receiver of the steganographic image will then decode to reveal the ciphertext of the message and then decrypt using an established shared key.

For the unsupervised design of steganographic techniques, we leverage ideas from the field of adversarial training [7]. Typically, adversarial training is used to train generative models on tasks such as image generation and speech synthesis. We design a scheme that aims to embed a secret message within an image. Our task is discriminative, the embedding algorithm takes in a *cover* image and produces a *steganographic* image, while the adversary tries to learn weaknesses in the embedding algorithm, resulting in the ability to distinguish cover images from steganographic images.

The success of a steganographic algorithm or a steganalysis technique over one another amounts to ability to model the cover distribution correctly [5]. So far, steganographic schemes have used human-based rules to 'learn' this distribution and perturb it in a way that disrupts it least. However, steganalysis techniques commonly use machine learning models to learn the differences in distributions between the cover and steganographic images. Based on this insight we pursue the following hypothesis:

**Hypothesis:** Machine learning is as capable as human-based rules for the task of modeling the cover distribution, and so naturally lends itself to the task of designing steganographic algorithms, as well as performing steganalysis.

In this paper, we introduce the first steganographic algorithm produced entirely in an unsupervised manner, through a novel adversarial training scheme. We show that our scheme can be successfully implemented in practice between two communicating parties, and additionally that with supervised training, the steganalyzer, Eve, can compete against state-of-the-art steganalysis methods. To the best of our knowledge, this is one of the first real-world applications of adversarial training, aside from traditional adversarial learning applications such as image generation tasks.

## 2 Related work

### 2.1 Adversarial learning

Two recent designs have applied adversarial training to cryptographic and steganographic problems. Abadi and Andersen [2] used adversarial training to teach two neural networks to encrypt a short message, that fools a discriminator. However, it is hard to offer an evaluation to show that the encryption scheme is computationally difficult to break, nor is there evidence that this encryption scheme is competitive against readily available public key encryption schemes. Adversarial training has also been applied to steganography [4], but in a different way to our scheme. Whereas we seek to train a model that learns a steganographic technique by itself, Volkhonskiy et al's. work augments the original GAN process to generate images which are more susceptible to established steganographic algorithms. In addition to the normal GAN discriminator, they introduce a steganalyzer that receives examples from the generator that may or may not contain secret messages. The generator learns to generate realistic images by fooling the discriminator of the GAN, and learns to be a secure container by fooling the steganalyzer. However, they do not measure performance against state-of-the-art steganographic techniques making it difficult to estimate the robustness of their scheme.

### 2.2 Steganography

Steganography research can be split into two subfields: the study of steganographic algorithms and the study of steganalyzers. Research into steganographic algorithms concentrates on finding methods to embed secret information within a medium while minimizing the perturbations within that medium. Steganalysis research seeks to discover methods to detect such perturbations. Steganalysis is a binary classification task: discovering whether or not secret information is present with a message, and so machine learning classifiers are commonly used as steganalyzers.

Least significant bit (LSB) [16] is a simple steganographic algorithm used to embed a secret message within a cover image. Each pixel in an image is made up of three RGB color channels (or one for grayscale images), and each color channel is represented by a number of bits. For example, it is

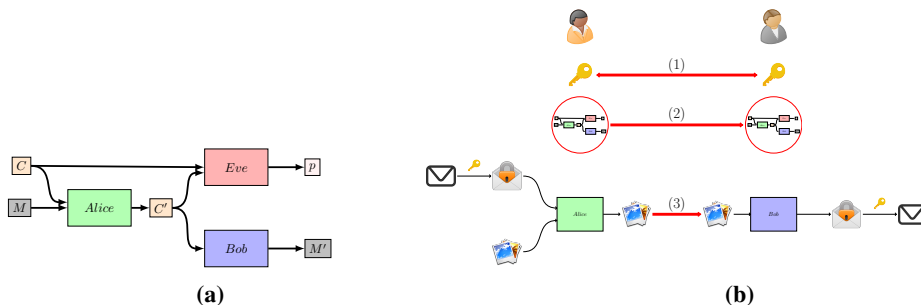

**Figure 1: (a)** Diagram of the training game. **(b)** How two parties, Carol and David, use the scheme in practice: (1) Two parties establish a shared key. (2) Carol trains the scheme on a set of images. Information about model weights, architecture and the set of images used for training is encrypted under the shared key and sent to David, who decrypts to create a local copy of the models. (3) Carol then uses the *Alice* model to embed a secret encrypted message, creating a steganographic image. This is sent to David, who uses the *Bob* model to decode the encrypted message and subsequently decrypt.

common to represent a pixel in a grayscale image with an 8-bit binary sequence. The LSB technique then replaces the least significant bits of the cover image by the bits of the secret message. By only manipulating the least significant bits of the cover image, the variation in color of the original image is minimized. However, information from the original image is always lost when using the LSB technique, and is known to be vulnerable to steganalysis [6].

Most steganographic schemes for images use a distortion function that forces the embedding process to be localized to parts of the image that are considered noisy or difficult to model. Advanced steganographic algorithms attempt to minimize the distortion function between a cover image, $C$, and a steganographic image, $C'$,

$$d(C, C') = f(C, C') \cdot |C - C'|$$

It is the choice of the function $f$, the cost of distorting a pixel, which changes for different steganographic algorithms.

HUGO [18] is considered to be one of the most secure steganographic techniques. It defines a distortion function domain by assigning costs to pixels based on the effect of embedding some information within a pixel, the space of pixels is condensed into a feature space using a weighted norm function. WOW (Wavelet Obtained Weights) [9] is another advanced steganographic method that embeds information into a cover image according to regions of complexity. If a region of an image is more texturally complex than another, the more pixel values within that region will be modified. Finally, S-UNIWARD [10] proposes a universal distortion function that is agnostic to the embedding domain. However, the end goal is much the same: to minimize this distortion function, and embed information in noisy regions or complex textures, avoiding smooth regions of the cover images. In Section 4.2, we compare out results against a state-of-the-art steganalyzer, ATS [13]. ATS uses labeled data to build artificial training sets of cover and steganographic images, and is trained using an SVM with a Gaussian kernel. They show that this technique outperforms other popular steganalysis tools.

## 3 Steganographic adversarial training

This section discusses our steganographic scheme, the models we use and the information each party wishes to conceal or reveal. After laying this theoretical groundwork, we present experiments supporting our claims.

### 3.1 Learning objectives

Our training scheme involves three parties: Alice, Bob and Eve. Alice sends a message to Bob, Eve can eavesdrop on the link between Alice and Bob and would like to discover if there is a secret message embedded within their communication. In classical steganography, Eve (the *Steganalyzer*) is passed both unaltered images, called *cover* images, and images with secret messages embedded

within, called *steganographic* images. Given an image, Eve places a confidence score of how likely this is a cover or steganographic image. Alice embeds a secret message within the cover image, producing a steganographic image, and passes this to Bob. Bob knows the embedding process and so can recover the message. In our scheme, Alice, Bob and Eve are neural networks. Alice is trained to learn to produce a steganographic image such that Bob can recover the secret message, and such that Eve can do no better than randomly guess if a sample is a cover or steganographic image.

The full scheme is depicted in Figure 1a: Alice receives a cover image, $C$, and a secret encrypted message, $M$, as inputs. Alice outputs a steganographic image, $C'$, which is given to both Bob and Eve. Bob outputs $M'$, the secret message he attempts to recover from $C'$. We say Bob performs perfectly if $M = M'$. In addition to the steganographic images, Eve also receives the cover images. Given an input $X$, Eve outputs the probability, $p$, that $X = C$. Alice tries to learn an embedding scheme such that Eve always outputs $p = \frac{1}{2}$. We do not train Eve to maximize her prediction error, since she can then simply flip her decision and perform with perfect classification accuracy. Figure 1b shows how the scheme should be used in pratice if two people wish to communicate a steganographic message using our scheme. The cost of sending the encrypted model information from Carol to David is low, with an average of 70MB. Note that in Figure 1b, steps (1) and (2), the set-up of the shared key and sharing of model information, is perfomed offline. We assume, as is common in cryptographic research, that this initial set-up phase is not visible to an adversary.

At the beginning of training, a human can easily separate cover images from steganographic images, as Alice has not learned yet how to embed the secret message such that there is no visible difference in the cover image. However, we train Eve much like a discriminator in a GAN, where we tie her predictive power to the embedding capacity of Alice. When Alice produces a steganographic image that does not resemble the cover image, Eve does not have the ability to perfectly separate cover from steganographic images. As training continues, Eve becomes better at her task, but then so does Alice as her weights are updated, in part, based on the loss of Eve.

Similarly to Abadi and Andersen [2], we let $\theta_A, \theta_B, \theta_C$ denote the parameters of Alice, Bob and Eve, respectively. We write $A(\theta_A, C, M)$ for Alice's output on $C$ and $M$, $B(\theta_b, C')$ for Bob's output on $C'$, and $E(\theta_E, C, C')$ for Eve's output on $C$ and $C'$. Let $L_A, L_B, L_C$ denote the loss of Alice, Bob and Eve, respectively. Then, we have the following relations:

$$B(\theta_b, C') = B(\theta_b, A(\theta_A, C, M))$$
$$E(\theta_E, C, C') = E(\theta_E, C, A(\theta_A, C, M))$$

We set Bob's loss (the secret message reconstruction loss), to be the Euclidean distance between $M$ and $M'$:

$$\begin{aligned} L_B(\theta_A, \theta_B, M, C) &= d(M, B(\theta_b, C')) \\ &= d(M, B(\theta_b, A(\theta_A, C, M))) \\ &= d(M, M') \end{aligned}$$

As is common with GAN discriminator implementations, we set the Eve's loss to be sigmoid cross entropy loss:

$$\begin{aligned} L_E(\theta_E, C, C') &= -y \cdot log(E(\theta_E, x)) \\ &\quad - (1 - y) \cdot log(1 - E(\theta_E, x)), \end{aligned}$$

where $y = 0$ if $x = C'$ and $y = 1$ if $x = C$. Alice's loss is given as a weighted sum of Bob's loss, Eve's loss on steganographic images, and an additional reconstructive loss term:

$$\begin{aligned} L_A(\theta_A, C, M) &= \lambda_A \cdot d(C, C') + \lambda_B \cdot L_B \\ &\quad + \lambda_E \cdot L_E(\theta_E, C, C'), \end{aligned}$$

where $d(C, C')$ is the Euclidean distance between the cover image and the steganographic image, and $\lambda_A, \lambda_B, \lambda_E \in \mathbb{R}$ define the weight given to each respective loss term.

Our goal is not only to explore whether neural networks can produce steganographic embedding algorithms in an unsupervised manner, but whether they are competitive against steganographic algorithms like HUGO, WOW and S-UNIWARD, that have been designed by steganography experts. We did not intend to encode a specific algorithm within the neural network, rather we would like to give the networks the opportunity to devise their own.

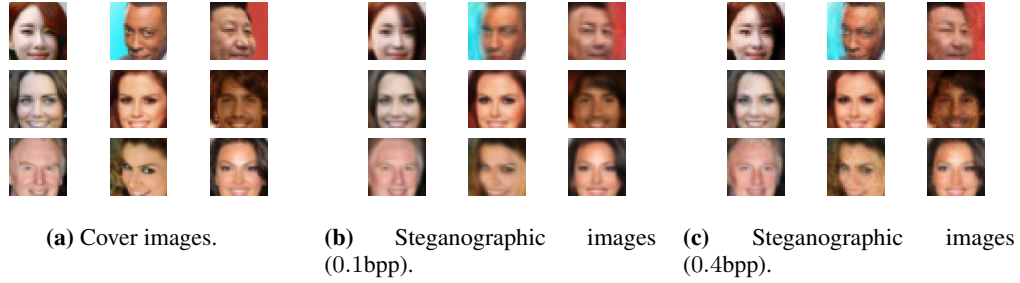

**(a)** Cover images.     **(b)** Steganographic images (0.1bpp).     **(c)** Steganographic images (0.4bpp).

**Figure 2:** Cover and steganographic images from the celebA dataset, with embedding rates of 0.1bpp and 0.4bpp.

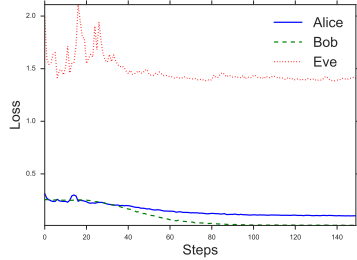

**(a)** Alice, Bob and Eve error for 0.1bpp.

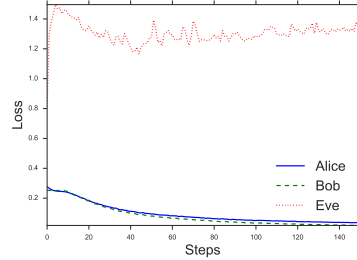

**(b)** Alice, Bob and Eve error for 0.4bpp.

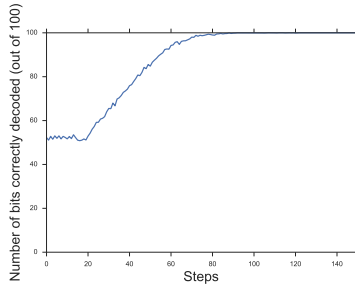

**(c)** Bob decoding success for 0.1bpp.

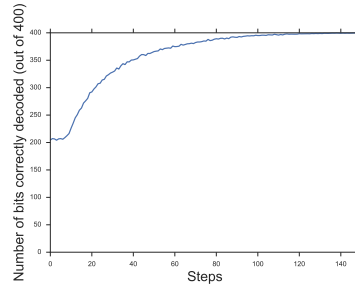

**(d)** Bob decoding success for 0.4bpp.

**Figure 3:** Results on celebA dataset at an embedding rate of 0.1bpp and 0.4bpp.

## 3.2 Architecture

**Alice (Steganographic Encoder).** Alice accepts as input a cover image and a random $n$-bit binary message. The cover image is flattened and concatenated with the secret message, which is then fed into the neural network. We would like the locality of the embedding process to be learned instead of fixed; the first layer is fully connected to the next, allowing the secret message to be combined with any region of the cover image. We then reshape to $4 \times 4 \times I \cdot 8$, where $I$ is width of the cover image. Each of the next four layers are a sequence of upsampling fractionally-strided convolutions, batch normalization and ReLU [17], except for the final layer where tanh [12] is used as the activation function.

**Bob (Steganographic Decoder).** Bob accepts as input the steganographic image output by Alice. Bob uses a deep convolutional architecture to extract the important bits from the output of Alice. It is made up of four fractionally-strided convolutions and one final fully connected layer. Leaky ReLU [15] is used in all but the final layer, where tanh is used. Batch normalization is applied at each layer.

**Eve (Steganalyzer).** Eve accepts as input both cover images and steganographic images. Eve has a similar architecture to Bob but uses the sigmoid activation function instead of tanh at the final layer to output probabilities from the logits.

# 4   Experiments

As a proof of concept, we implemented our adversarial training scheme on two image datasets: celebrity faces in the wild (celebA) [14] and a standard steganography research dataset, BOSS[2]. We concatenate an $n$-bit random message, with each sample of each dataset. We vary the size of the message, $n$, to test the limits of the amount of information that can be effectively hidden inside a cover image. This corresponds to the number of bits per pixel (bpp) that can be hidden by a steganographic algorithm, a common measure of the success of a steganographic research. On average, steganographic algorithms can successfully hide approximately $0.4$bpp. Our datasets are formed of $32 \times 32$ pixel images, and so we vary $n$ from $100$ to $400$, this corresponds to hiding between approximately $0.1$bpp to $0.4$bpp. We intentionally randomize the secret message so that the network does not learn a specific function associated with a specific group of messages. We want to be sure that our model can generalize to any message that is less than $n$ bits[3]. For each experiment, we performed grid search to find the optimum loss weights, $\lambda_A, \lambda_B, \lambda_E$, for Alice. All experiments in this section were performed in TensorFlow [1, 3], on a workstation with a Tesla K40 GPU card.

## 4.1   CelebA dataset

We first motivate our work by showing our model can produce steganographic images that are almost indistinguishable from the cover images. We vary the size of the embedded secret message, to explore how many bits may be stored in a cover image while remaining undetected by a steganalyzer. As expected, we find that a larger perturbation in the cover image results in more communicated bits but carries a greater risk of detection.

The celebA dataset consists of $202,599$ images of celebrity faces from $10,177$ unique identities. We train in batches of 32, and use the Adam optimizer [11] with a learning rate of $2 \cdot 10^{-4}$. At each batch we alternate training either Alice and Bob, or Eve.

Figure 2b and Figure 2c show the steganographic images output by Alice after 150 training steps, for embedding rates of $0.1$bpp and $0.4$bpp, respectively. Our scheme easily copes with an embedding rate of $0.1$bpp for a $32 \times 32$ image, which is equivalent to communicating a short 12 character ASCII message. Figure 3a and Figure 3c shows the training loss of each model, and the success of Bob decoding the secret message over time, respectively. In the first few rounds of training, the visual quality of Alice's output is low and Bob is essentially randomly guessing the message. After approximately 30 steps, Alice learns to successfully embed information in a cover image such that Eve is fooled, and Bob learns to correctly decode the message. After 150 training steps, Bob decodes the message with an average success of 100%, while Eve's loss remains the same. Figure 3b and Figure 3d show similar results, with both sample quality and success of decoding both high by the end of training. Concealing $0.4$bpp corresponds to successfully hiding a message of 51 ASCII characters. Due to the larger amount of information communicated between Alice and Bob, it takes a longer period of time for Alice to discover an effective embedding scheme, and for Bob to correctly discover the secret message. After 150 training steps, Bob can successfully decode 100% of the message.

There is no noticeable image quality decrease between $0.1$bpp and $0.4$bpp, however we found after $0.4$bpp, image quality suffered and it was no longer possible to balance realistic image creation capable of fooling Eve, and convergence of message decoding. For both embedding rates of $0.1$bpp and $0.4$bpp, Eve's loss remained constant at approximately $1.4$. Eve's loss is the sum of losses on both steganographic images and cover images, which both remained steady at approximately $0.69$. Due to the logarithmic term in the loss function, this is equivalent to randomly guessing if an image carries a secret message.

## 4.2   BOSS dataset

In addition to our experiments on the celebA dataset, we trained our steganographic scheme on the BOSS image dataset, which is commonly used as a benchmark in steganography research. BOSS is made up of $10,000$ grayscale images depicting a variety of scenes such as landscapes, buildings and animals. We expected our scheme to perform worse than on the celebA dataset, since the cover

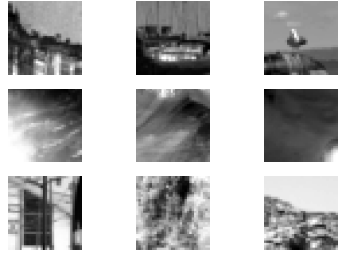

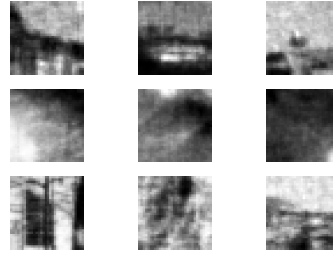

(a) Cover images of buildings, birds, skies and the ocean.

(b) Steganographic images (0.1bpp).

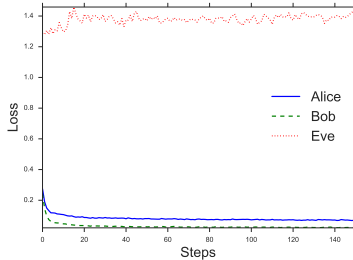

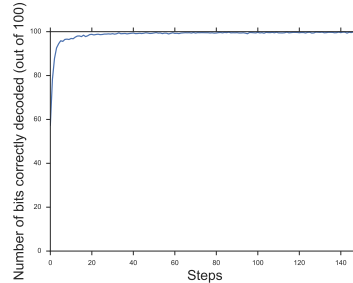

(c) Alice, Bob and Eve error for 0.1bpp.

(d) Bob decoding success for 0.1bpp.

**Figure 4:** Results on BOSS dataset at an embedding rate of 0.1bpp.

images do not come from a single distribution. However, we found our scheme is still capable of embedding secret information successfully.

Figure 4b shows the sample quality of steganographic images with an embedding rate of 0.1bpp, while Figure 4c and Figure 4d show the error rates of Alice, Bob and Eve, and the success of Bob decoding the secret message, respectively. While image quality suffers slightly more than on the celebA dataset, our scheme is still able to learn a steganographic algorithm. Our scheme is output samples that are not dissimilar from the original dataset, while Bob is able to learn to successfully decode the message. Alice and Bob both learn their respective tasks in a relatively short period of time, after which there is not much improvement in terms of hiding or recovering the secret message. At the end of training, Bob is able to successfully decode the secret message with 99.8% accuracy.

## 4.3 Comparison with related work

Fooling a steganalyzer, Eve, is easy by design, since we train in such a way that Eve never has a significant competitive advantage. Thus, we additionally show that the resultant trained steganographic model, Alice, can fool an independent steganalyzer. We compare our scheme against both state-of-the-art steganographic algorithms and steganalysis tools and show that it performs competitively.

For both BOSS and CelebA, we compare our scheme against steganographic algorithms HUGO, WOW and S-UNIWARD. Additionally, we implement the Eve model using supervised training and compare against the steganalyzer ATS in Table 1. By design, Eve only performs slightly better than random. One may wonder whether the choice of model for Eve is wise; why not use an established steganalyzer in place of the Eve model? By training Eve in a supervised fashion, we show that Eve has the capacity to become a strong steganalyzer, competing against established techniques like ATS, and so is a good choice for the steganalyzer. Furthermore, Eve does not require a feature extraction preprocessing step as with ATS, and, from our experiments, is an order of magnitude quicker to train. For both the BOSS and CelebA datasets, we use $10,000$ samples and split in half, creating a training set and a test set. Alice was then trained on the 5000 samples from the training set. We then created an additional $10,000$ steganographic images for each steganographic algorithm (Alice, HUGO, WOW and S-UNIWARD). Now each steganographic algorithm has an associated training

**Table 1:** Accuracy of distinguishing between cover and steganographic images for the steganalyzers, Eve and ATS, on the BOSS and CelebA datasets at an embedding rate of 0.4bpp.

|  | BOSS | | CELEBA | |
| STEGANOGRAPHIC ALGORITHM | STEGANALYZER | | STEGANALYZER | |
|  | ATS | EVE | ATS | EVE |
| ALICE | 0.83 | 0.79 | 0.95 | 0.90 |
| HUGO | 0.66 | 0.59 | 0.94 | 0.89 |
| WOW | 0.75 | 0.74 | 0.89 | 0.85 |
| S-UNIWARD | 0.77 | 0.72 | 0.91 | 0.84 |

set and test set, each consisting of 5000 cover images and 5000 steganographic images. For each steganographic algorithm we train both ATS and Eve on the associated training set, and then report accuracy of the steganalyzer on the test set. From Table 1, Eve performs competitively against the steganalyzer, ATS, and Alice also performs well against other steganographic techniques. While our scheme does not substantially improve on current popular steganographic methods, it is clear that it does not perform significantly worse, and that unsupervised training methods are capable of competing with expert domain knowledge.

## 4.4 Evaluating robust decryption

Due to the non-convexity of the models in the training scheme, we cannot guarantee that two separate parties training on the same images will converge to the same model weights, and so learn the same embedding and decoding algorithms. Thus, prior to steganographic communication, we require one of the communicating parties to train the scheme locally, encrypt model information and pass it to the other party along with information about the set of training images. This ensures both parties learn the same model weights. To validate the practicality of our idea, we trained the scheme locally (Machine A) and then sent model information to another workstation (Machine B) that reconstructed the learned models. We then passed steganographic images, embedded by the *Alice* model from Machine A, to Machine B, who used the *Bob* model to recover the secret messages. Using messages of length corresponding to hiding 0.1bpp, and randomly selecting 10% of the CelebA dataset, Machine B was able to recover 99.1% of messages sent by Machine A, over 100 trials; our scheme can successfully decode the secret encrypted message from the steganographic image. Note that our scheme does not require perfect decoding accuracy to subsequently decrypt the message. A receiver of a steganographic message can successfully decode and decrypt the secret message if the mode of encryption can tolerate errors. For example, using a stream cipher such as AES-CTR guarantees that incorrectly decoded bits will not affect the ability to decrypt the rest of the message.

## 5 Discussion & conclusion

We have offered substantial evidence that our hypothesis is correct and machine learning can be used effectively for both steganalysis and steganographic algorithm design. In particular, it is competitive against designs using human-based rules. By leveraging adversarial training games, we confirm that neural networks are able to discover steganographic algorithms, and furthermore, these steganographic algorithms perform well against state-of-the-art techniques. Our scheme does not require domain knowledge for designing steganographic schemes. We model the attacker as another neural network and show that this attacker has enough expressivity to perform well against a state-of-the-art steganalyzer.

We expect this work to lead to fruitful avenues of further research. Finding the balance between cover image reconstruction loss, Bob's loss and Eve's loss to discover an effective embedding scheme is currently done via grid search, which is a time consuming process. Discovering a more refined method would greatly improve the efficiency of the training process. Indeed, discovering a method to quickly check whether the cover image has the capacity to accept a secret message would be a great improvement over the trial-and-error approach currently implemented. It also became clear that Alice and Bob learn their tasks after a relatively small number of training steps, further research is needed to explore if Alice and Bob fail to improve due to limitations in the model or because of shortcomings in the training scheme.

# 6 Acknowledgements

The authors would like to acknowledge financial support from the UK Government Communications Headquarters (GCHQ), as part of University College London's status as a recognised Academic Centre of Excellence in Cyber Security Research. Jamie Hayes is supported by a Google PhD Fellowship in Machine Learning. We thank the anonymous reviewers for their comments.

## Footnotes

[1]See EFF's guide: https://www.eff.org/files/2014/05/29/unnecessary_and_disproportionate.pdf.

[2]`http://agents.fel.cvut.cz/boss/index.php?mode=VIEW&tmpl=materials`

[3]This ensures our scheme can embed ciphertexts of messages, which appear as random strings.

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
