[Reviews · NeurIPS 2017]

Reviewer 1



The authors studied how to use adversarial training to learn the encoder, the steganalyzer and the decoder at the same time using unsupervised learning. Specifically, the authors designed an adversarial game of three parties. The encoder generates images based on the cover and the message. Then the generated image can be received to both decoder and steganalyzer. The goal of the encoder and decoder is to correctly encode and decode the message, and the goal of the steganalyzer is to determine whether the image is encrypted. Thus it is a minimax game. Despite the algorithmic part of the paper first, my major concern is the correctness of the methodology. As we all know, the neural networks have great ability to fit or classify the training data. You use two almost same neural network architectures as the decoder and steganalyzer. Then the question raises, as the decoder and steganalyzer use the same information in the training process. If the steganalyzer can correctly predict the label of the image, then the task fails. But if the steganalyzer cannot correctly predict the label, I don't think the decoder can recover the message well. I am confused about this and hope the author can answer the question in the rebuttal phase. From the experiment we can see the loss of steganalyzer is super large according to figure 3 and 4, which means that the steganalyzer fails to classify almost all examples. However, the authors also shown that steganalyzer is "pretty well" if the steganographic algorithm is "fixed". I am also confused about such self-contradictory explanations. Do you mean that during training, your steganalyzer is not well learnt? And your encoder and decoder is learnt using a bad steganalyzer? Here are also some minor comments: 1. Line 151: the loss function L_E has no dependency on \theta_A given C' 2. Line 152: parameter C should be included into loss function L_E 3. A large number of details are missed in the experiments. What are the values of hyperparameters? In the paper, the author claimed it is converged after 150 training steps. As the batch size is 32, you cannot even make a full pass to the training examples during training. How can you explain this?

Reviewer 2



This work presents a straightforward and exciting application of adversarial training to steganography: One network is trained to conceal a message in an image while a second one (the steganalyser) is trained to tell whether a given image contains a secret message or not. The framework is well suited to adversarial training, the paper is well written, and the necessary material relative to security is carefully introduced. The adversarial steganography based technique is compared to standard handcrafted methods using public datasets. The experiments show the superiority of the presented approach. My primary concern with this paper is the experimental setting: In practice, one always uses several steganalysers rather than just one. Besides, the quality of the steganographer is tied to the capacity of the steganalyser. Therefore, a stronger steganalyser will likely be able to tell legitimate images apart from altered ones. I understand this problem is more general to adversarial training and goes beyond this particular application. Nonetheless, its importance is particularly emphasized in this case and calls for further investigation. Despite this remark, I believe this is an exciting contribution that opens a new avenue for application of adversarial training to steganography. The originality of the work, the importance of the problem and the clarity of the presentation outweighs the previously mentioned point. I vote for acceptance.

Reviewer 3



The paper presents a very important interesting case study that existing machine learning algorithms can be used for a real application of adversarial training. We general, we like the paper, but feel that the venue is more of a case study and and the novelty of the work is not currently high or impactful enough to for the wider NIPS community. We would suggest other venues, unless the novelty of the main idea of the paper develop more strongly.

Reviewer 4



This manuscript proposes to use GANs to learn a steganographic algorithm in an unsupervised manner. This is modeled with three neural networks: Alice, Bob and Eve. Alice produces images that include or do not include embedded messages. Bob is the intended recipient and decodes the information from these images. Eve is a malicious entity that attempts to determine whether a message was embedded or not in a "cover" picture. In Section 3.1, Eve is assumed to have access to both the original image and the cover image. It might also make sense to also consider the case where Eve has access to samples drawn from the distribution of cover images, but not necessarily the specific cover image which was used by Alice to embed a message. It would also be interesting to study the impact of the capacity of Eve's neural network on its ability to "win" the adversarial training game. The evaluation would be stronger if augmented with a comparison of different model architectures for the three neural networks involved. Finally, perhaps the weakest point of this manuscript is the absence of a proper steganalysis. This is an indirect consequence of the lack of interpretability of neural networks, but should be at least acknowledged in the writing. This manuscript is well written. Here are some minor comments: - lines 135-136: perhaps mention public key cryptography as a means to achieve this, even on public communication channels. - lines 140 -142: I am not sure I understood what was meant here. If Alice is not able to produce steganographic images that resemble the cover images, shouldn't Eve have an easy time identifying the steganographic images? - lines 144 -147: the notation definitions should be repeated here to make it easier for the reader to parse the output of the different neural networks. - Section 4.2: is there a particular reason for not including the encoding/decoding success for 0.4bpp? Is it because the dimensionality of the images does not allow for embedding so many bits per pixels?